# The Impact of Hydroxychloroquine on Primary Feto-Placental Endothelial Cells from Healthy and Early-Onset Preeclamptic Placentas

**DOI:** 10.3390/ijms241310934

**Published:** 2023-06-30

**Authors:** Maja Gajić, Bianca Schröder-Heurich, Monika Horvat Mercnik, Mila Cervar-Zivkovic, Christian Wadsack, Frauke von Versen-Höynck, Karoline Mayer-Pickel

**Affiliations:** 1Department of Obstetrics and Gynecology, Medical University of Graz, Auenbruggerplatz 14, A-8036 Graz, Austria; maja.gajic@medunigraz.at (M.G.); monika.horvat@medunigraz.at (M.H.M.); mila.cervarzivkovic@medunigraz.at (M.C.-Z.); christian.wadsack@medunigraz.at (C.W.); 2Gynecology Research Unit, Hannover Medical School, Carl-Neuberg-Strasse 1, D-30625 Hannover, Germany; schroeder-heurich.bianca@mh-hannover.de (B.S.-H.); vonversen-hoeynck.frauke@mh-hannover.de (F.v.V.-H.); 3BioTechMed-Graz, Mozartgasse 12/II, A-8010 Graz, Austria; 4Department of Obstetrics and Gynecology, Hannover Medical School, Carl-Neuberg-Strasse 1, D-30625 Hannover, Germany

**Keywords:** hydroxychloroquine (HCQ), preeclampsia (PE), endothelial cells, VCAM-1, IL-8

## Abstract

Hydroxychloroquine (HCQ), an anti-malarial drug, is suggested as a promising candidate for the treatment of pregnancy-related disorders associated with endothelial activation, among which there is preeclampsia (PE). Arterial feto-placental endothelial cells (fpECAs) were isolated from control (CTR) and early-onset preeclamptic (EO-PE) placentas. The aim of this study was to test potential protective effects of HCQ in an in vitro model of endothelial activation as well as in cells isolated from EO-PE placentas. To mimic PE conditions, CTR fpECAs were exposed to a pro-inflammatory environment consisting of tumor necrosis factor α (TNF-α), interleukin (IL)-6 and IL-1β (furtherly referred as MIX) with or without varying concentrations of HCQ (1 µg/mL and 10 µg/mL). Their effect on wound healing and endothelial barrier integrity was analyzed. Variations in the expression of IL-8 and leukocyte adhesion molecules (LAM) on both mRNA and protein levels were determined between CTR and PE fpECAs in the presence or absence of HCQ. MIX decreased wound healing and stability of the endothelial barrier, but HCQ did not affect it. Significant differences between CTR and EO-PE fpECAs were observed in IL-8 mRNA, protein secretion, and *vascular cell adhesion protein 1* (*VCAM-1)* mRNA expression levels. After challenging CTR fpECAs with MIX, upregulation of both mRNA and protein levels was observed in all molecules. Combined treatment of HCQ and MIX slightly lowered VCAM-1 total protein amount. In CTR fpECAs, treatment with low concentrations of HCQ alone (1 µg/mL) reduced basal levels of *IL-8* and *VCAM-1* mRNA and secretion of IL-8, while in EO-PE fpECAs, a higher (10µg/mL) HCQ concentration slightly reduced the gene expression of *IL-8*. Conclusion: These results provide additional support for the safety of HCQ, as it did not adversely affect endothelial functionality in control fpECAs at the tested concentration. Furthermore, the observed limited effects on IL-8 secretion in EO-PE fpECAs warrant further investigation, highlighting the need for clinical trials to assess the potential therapeutic effects of HCQ in preeclampsia. Conducting clinical trials would offer a more comprehensive understanding of HCQ’s efficacy and safety, allowing us to explore its potential benefits and limitations in a real-world clinical setting.

## 1. Introduction

Hydroxychloroquine (HCQ), originally an anti-malarial drug, is currently used for the treatment of different autoimmune diseases, such as systemic lupus erythematosus [1], anti-phospholipid syndrome [2], and Sjögren’s syndrome [3,4,5], mostly due to its various anti-inflammatory and immune-regulatory abilities. There are a few possible mechanisms of action of HCQ described by various authors [6,7]. It has been shown that HCQ blocks the production of several pro-inflammatory cytokines such as interleukin (IL)-17, IL-6, interferon γ, and tumor necrosis factor α (TNF-α) [8]. In contrast, it increases the production of anti-inflammatory cytokines IL-10 and IL-16 and shifts the immune response toward the Th2 pattern [9]. HCQ is lyophilic and can easily pass through the cell membrane, where it accumulates in acidic lysosomes, affecting their pH. Therefore, HCQ prevents lysosomal enzyme activation [6], and inhibits endosomal toll-like receptors [10]. Through these functions, HCQ might prevent systemic inflammation. While some studies could not confirm a difference in preeclampsia (PE) incidence within pregnant lupus patients who continued taking HCQ treatment during pregnancy [11], others have shown an association with a lower risk of PE and HCQ intake [12] and higher gestational age at delivery [13]. Moreover, it has been demonstrated that HCQ crosses the placenta [14], and its safety during pregnancy has been confirmed in a few studies [10,11,15]. Some studies suggest that HCQ may have beneficial effects on endothelial function by reducing inflammation and by improving vascular health [16]. For this reason, HCQ is proposed as a promising candidate for the treatment of pregnancy-related disorders related to endothelial activation, including PE [17,18].

We postulate hypotheses that HCQ would protect endothelial cells from cytokine storm and lower levels of molecules connected to endothelial activation. We explore this hypothesis through three aims of this study. The first aim was to establish an in vitro model of cytokine-induced endothelial activation that mimics the process observed during PE. The second aspect focused on comparing the levels of inflammation-related molecules between feto-placental endothelial cells isolated from arteries (fpECAs) derived from normal (CTR) and early-onset preeclamptic (EO-PE) placentas. The third aim involved investigating the potential protective effects of HCQ in an established in vitro model of endothelial activation as well as the impact of HCQ on the baseline state of fpECAs derived from both CTR and EO-PE placentas.

## 2. Results

### 2.1. Patients Data

Characteristics from the anonymized pregnant women and their neonates were obtained and are reported in Table 1 as mean ± standard deviation (SD). Cells isolated from 14 CTR and 7 EO-PE placentas were used for this study. When characteristics of patients were compared (Table 1), differences were observed in gestational age (*p* < 0.001), fetal birth weight (*p* < 0.001) and percentile (*p* = 0.005). Maternal age, BMI before pregnancy and pH of umbilical cord blood from either artery or vein were comparable.

### 2.2. Receptor Expression Analysis of TNF-α, IL-6, and IL-1β in fpECAs

Prior to cytokine stimulation, the presence of respective receptors in fpECAs was investigated. It was important to test for their presence since it is not clear if cytokines from maternal sites passing through the placenta as well as how much inflammation happens on fetal side. Moreover, levels of receptors for TNFα, IL-6, and IL-1β were taken in consideration for interpretation of the obtained results. Similar receptor levels would suggest alternative mechanisms driving the differences in the expression of inflammatory molecules, while differing receptor levels could indicate varying cellular responses to cytokine stimulation. Messenger RNA (mRNA) expression of TNF-α, IL-6 and IL-1β receptors was determined by quantitative reverse transcription PCR (RT-qPCR) (Figure 1A–C). Expression of all three receptors was determined in both CTR and EO-PE fpECAs. Moreover, it was observed that the expression of all three cytokine receptors is comparable between the groups (Figure 1A–C).

### 2.3. Impact of HCQ and MIX on Endothelial Functionality of CTR fpECAs

Endothelial dysfunction, present in multiple pregnancy systemic disorders, including PE, is reflected by a disturbed endothelial barrier, decreased migration and decreased proliferation of endothelial cells. To explore the effects of 1 µg/mL HCQ and a cytokine mix composed of TNF-α, IL-6 and IL-1β (in further text referred to as MIX) on endothelial functionality, we conducted experiments at different time points in CTR fpECAs. The wound healing rate was highly suppressed by both MIX (with and without HCQ) after 15 h (*p* = 0.0018, Figure 2B.III), while no differences were observed after 5 h (Figure 2B.I) and 10 h (Figure 2B.II), compared to CTR. Notably, the addition of HCQ alone or in combination with MIX improved wound closure by 5% at the 15 h time point, although it did not reach statistical significance (Figure 2B.III).

Stability of the endothelial barrier of fpECAs was tested by continuously measuring impedance values at time intervals of 15 min right after the addition of HCQ and MIX. The cell index of HCQ-treated fpECAs did not differ from untreated fpECAs (CTR). However, the MIX treatment of CTR fpECAs results in disruption of the endothelial barrier, and it was the most prominent after 5 h (*p* = 0.01, Figure 3B.I). Over time, the endothelial barrier slowly recovered, as confirmed statistically at 10 h (*p* = 0.009, Figure 3B.II) and 15 h (*p* = 0.028, Figure 3B.III), but did not reach CTR levels. fpECAs incubated with both MIX and HCQ exhibited a similar breakdown of the endothelial barrier compared to MIX-treated fpECAs, indicating that HCQ did not protect endothelial barrier integrity (Figure 3A).

### 2.4. Evaluation of Endothelial Activation Markers in fpECAs Isolated from CTR and EO-PE Placentas: Focus on IL-8, ICAM-1, VCAM-1 and Selectin E 

Because TNFα, IL-6 and IL-1β were used as the challenge in our in vitro cellular fpECAs model, we used different molecules as a readout for endothelial activation. For this purpose, mRNA and protein levels of IL-8 and leukocyte adhesion molecules (LAM) were chosen. Variations of LAM on both mRNA and protein expression levels was computed between CTR and PE fpECAs with RT-qPCR and flow cytometry, respectively. mRNA (Figure 4A.I) and membrane abundance (Figure 4A.II) of intercellular adhesion molecule 1 (ICAM-1) did not differ between CTR and EO-PE fpECAs. *Selectin E* (*SELE*) mRNA levels were comparable between CTR and PE, (Figure 4B.I) while membrane protein amount was higher in EO-PE fpECAs (*p* = 0.029, Figure 4B.II). In EO-PE fpECAs, expression of *vascular cell adhesion protein 1* (*VCAM-1)* (*p* = 0.004, Figure 4C.I) mRNA (I) was found to be upregulated compared to CTR fpECAs, which was also detectable on the membrane in the presence of VCAM-1 (*p* = 0.033, Figure 4C.II). In addition, a significant increase was observed on mRNA (*p* = 0.0024, Figure 4D.I) and secretory (measured by ELISA) (*p* = 0.0164, Figure 4D.II) levels of IL-8 in EO-PE fpECAs compared to CTR fpECAs.

### 2.5. Protective Effect of HCQ in an In Vitro Model of Endothelial Activation: Modulation of VCAM-1 and IL-8 Expression

The first aim of our study was to determine the potential protective effect of 1 µg/mL HCQ in an in vitro model of endothelial activation. In order to do that, we challenged CTR fpECAs with MIX alone or in combination with HCQ. Cells were also incubated with HCQ alone. In CTR fpECAs, MIX induced an increase in *ICAM-1* (Appendix A; Appendix A), *SELE* (Appendix A; Appendix A) and VCAM-1 (*p* ≤ 0.001, Figure 5A.I) mRNA. It also increased the membrane presence of *ICAM-1* (Appendix A; Appendix A) and VCAM-1 (*p* = 0.003, Figure 5A.II), but not SELE (Appendix A; Appendix A). However, when CTR cells were treated with HCQ alone, there was a significant downregulation only of *VCAM-1* mRNA (*p* ≤ 0.05, Figure 5A.I,). The VCAM-1 signal was only detectable for MIX analyzed by ELISA (Figure 5A.III) and Western blot (WB) (Figure 5A.IV). More importantly, a significant reduction of VCAM-1 protein expression (*p* = 0.01, Figure 5A.V) was observed in samples that were additionally treated with HCQ. This decrease was not led by changes in transcriptional (I), membrane (II) or secretion levels (III) of VCAM-1.

CTR fpECAs basal levels of IL-8 mRNA (Figure 5B.I) and protein secretion (Figure 5B.II) were affected by both HCQ and MIX. While HCQ significantly diminished IL-8 mRNA (*p* ≤ 0.05, Figure 5B.I) and secretory levels (*p* ≤ 0.01, Figure 5B.II) compared to untreated cells, MIX led to the activation of CTR fpECAs by increasing mRNA levels and significantly boosted the secretion of IL-8 (*p* ≤ 0.001, Figure 5B.II).

### 2.6. Dose-Dependent Changes in Basal Expression of IL-8 and VCAM in EO-PE fpECAs 

Since EO-PE fpECAs were directly isolated from six preeclamptic placentas, which were under an inflamed state due to the underlying pathology, they were incubated only with HCQ. We detected a difference between CTR and EO-PE fpECAs in VCAM-1 and IL-8; we next tested the mRNA expression of both of them after HCQ treatment. Moreover, membrane presence and secretion of VCAM-1 and IL-8 in HCQ-treated EO-PE fpECAs were measured. Besides a concentration of 1 µg/mL, it was decided to also treat EO-PE fpECAs with 10 µg/mL HCQ. Although 1 µg/mL had an effect on CTR fpECAs *IL-8* and *VCAM-1* mRNA levels, it did not induce any changes on transcriptional and translational levels of these molecules in EO-PE fpECAs (Figure 6). In addition, the higher concentration of HCQ affected either mRNA (Figure 6A.I) or membrane (Figure 6A.II) amounts of VCAM-1 in EO-PE fpECAs. However, this higher HCQ concentration reduced *IL-8* mRNA expression (*p* ≤ 0.05, Figure 6.BI) while it did not affect IL-8 secretion levels (Figure 6.BII).

## 3. Discussion

This study focused on the exploration of HCQ effects with regard to cytokine-induced endothelial activation in feto-placental endothelial cells, addressing three main aspects. The first one was to set in an in vitro model of endothelial activation similar to one with PE. Next, the second aspect was to assess the difference in inflammation-related molecules between fpECAs from CTR and EO-PE placentas. The third aim was to determine the potential protective effect of HCQ in an in vitro model of endothelial activation, as well as the effect of HCQ on the basal state of fpECAs from both CTR and EO-PE placentas. Since TNF-α, IL-6 and IL-1β are well-known inducers of endothelial activation, the use of all of them as a challenge might better reflect PE conditions in vitro instead of TNF-α alone. The increase in systemic TNF-α levels during pregnancy is associated with miscarriages, late fetal losses, PE, and preterm birth [19]. It also inhibits trophoblast-derived JEG-3 cell line integration into endothelial cellular networks in in vitro co-cultures with uterine-derived endothelial cells [20], implicating possible effects on first-trimester trophoblast invasion and remodeling of spiral arteries. Thus, TNF-α acts as one major mediator in the pathogenesis of severe PE [8]. In response to various stimuli, many different cell types produce IL-6, including endothelial cells [21]. It is also significantly increased in patients with pregnancy-induced hypertension [22,23] and in EO-PE [24] compared to normotensive pregnant women. In addition to TNF-α and IL-6, IL-1β was significantly increased in the sera as well as in the placentas of women with PE compared to healthy women [23,25,26].

First, challenging fpECAs with MIX, we determined the presence of respective cytokine receptors in CTR and EO-PE fpECAs by RT-qPCR. It was important to test for their presence since it is not clear if cytokines from the maternal site pass the placenta. Moreover, the amount of inflammation occurring on the fetal side is unknown. Differences in the expression of TNFα, IL-6, and IL-1β would affect the interpretation of IL-8 and VCAM-1 results. If the levels of receptors for TNFα, IL-6 and IL-1β were similar between CTL and EO-PE cells, it would suggest that any variations observed in IL-8 and VCAM-1 expression are more likely attributed to downstream signaling events or differential regulation at the mRNA and protein levels. In this case, altered intracellular signaling pathways, transcriptional regulation, or post-transcriptional modifications could primarily drive the differences in IL-8 and VCAM-1 expression specific to EO-PE cells.

On the other hand, if the levels of receptors for TNFα, IL-6, and IL-1β differ between CTL and EO-PE cells, it suggests potential differences in the cellular response to cytokine stimulation. A higher expression of these receptors in EO-PE cells could indicate an in-creased sensitivity or heightened inflammatory response to the pro-inflammatory cyto-kines. This, in turn, could contribute to the observed differences in IL-8 and VCAM-1 ex-pression between the two cell types.

mRNA expression of all three receptors was unchanged in CTR and EO-PE fpECAs. Therefore, IL-8 and VCAM-1 expression are likely influenced by downstream signaling events, mRNA and protein regulation, or post-transcriptional modifications specific to EO-PE cells. The effect of HCQ on the expression of these receptors was not tested, since it has been shown that HCQ targets downstream molecules in TNF-α, IL-6 and IL-1β path-ways, more specifically NFkB, p38, JNK [27] and ERK5 [28]. Hence, we expect that the observed HCQ effect is independent of a number of receptors.

Systematic endothelial dysfunction is present in women with PE [27,28]. The primary objective of this experiment was to assess the functional changes of CTR fpECAs under the influence of the cytokine mix, specifically focusing on wound healing and endothelial barrier integrity. By examining the endothelial barrier in CTR fpECAs, we aimed to establish a baseline understanding of the effects of the cytokine mix on endothelial functionality in a controlled setting. Similar to PE, cytokine MIX led to a decrease in wound healing capacity and endothelial barrier integrity of fpECAs. However, the addition of HCQ did not affect it. Consistent with our results from fpECAs, Dong et al. showed that HCQ neither improved nor impaired wound healing capacities of human umbilical vein endothelial cells (HUVEC) [29]. Rezabakhsh et al. showed that HCQ improves the high-glucose-mitigated wound healing process in HUVEC [30], while Ma et al., in 2017, did not observe any significant differences between CTR and HCQ in fibroblast-like synoviocytes [31].

Our results regarding the differences in IL-8 and LAM between CTR and EO-PE fpECAs on transcriptional and protein levels are in contrast to the data from other studies [32,33]. In those studies, the authors computed the differences between HUVEC isolated from healthy and PE placentas. Therefore, the contradictory results could be explained by the different cell types used in the studies. Both fpECAs and HUVEC exhibit typical endothelial molecular markers and they are both of fetal origin. However, they are isolated from vessels with opposite functions. Primary HUVEC originate from the umbilical cord vein, while fpECAs are isolated from arterial vessels of the placental chorionic plate. Moreover, Lang et al., in 2003, showed that fpECAs are good representatives of differentiated endothelial cells. It is stated that fpECAs share many features with adult arterial endothelial cells [34]. Hence, they represent a more reliable model for PE-induced endothelial dysfunction, since the maternal endothelium is more affected in PE [35].

IL-8 is a prominent chemokine involved in multiple processes such as acute inflammation [36,37] and angiogenesis [38]. Additionally, IL-8 was found to be increased in the serum and plasma of PE compared to healthy pregnancies [23,39,40]. In the present study, IL-8 was slightly reduced by HCQ on both mRNA and secretory levels in CTR fpECAs. However, cytokine-induced elevation of IL-8 was not attenuated by HCQ. These results indicate that the impact of HCQ on IL-8 is independent of the cytokine cascade. The results of this study are contradictory to the findings by Deleuran et al. [41]; the authors could not reveal any changes in IL-8 secretion in HUVEC stimulated by HCQ. An explanation might be in the difference and sensitivity of the methods used as a readout or again by the different cell types used. There is conflicting data regarding the effect of HCQ on serum levels of IL-8. While Wakiya et al. [42] showed a negative effect of HCQ, others did not observe any changes [43,44]. Therefore, it remains unresolved to what extent the levels of these cytokine relate to the severity of PE and its phenotypes [45].

Another possible mechanism of HCQ, as a potential treatment of PE, is through the NFκB signaling pathway. It has been shown that NFκB is one of the major regulators of leukocyte rolling and adhesion on the gene level [46]. The most important molecules in this process are ICAM-1, VCAM-1 and selectins [47]. When endothelial cells undergo inflammatory activation, they increase the expression of selectins, VCAM-1, and ICAM-1. In turn, they promote monocyte adherence and transfer to sites of inflammation. In our setup, we observed that MIX led to a rise in mRNA levels of all three tested molecules, but HCQ affected only *VCAM-1*. In line with several other studies [48,49], HCQ induced a decline in *VCAM-1*. It has been suggested that HCQ inhibits *VCAM-1* expression through the activation of ERK5 [48]. Similar to Kadife et al., in 2022, in HUVEC [50], we observed in fpECAs that HCQ led to a reduction in *VCAM-1*. However, in our set up, this effect was higher on total protein than on mRNA levels of VCAM-1. Discrepant results between mRNA and protein levels are frequently observed in biological studies and can be attributed to various factors. The levels of mRNA primarily determine the protein amounts at steady state. During transition to the steady state, there is a delayed synthesis between mRNA and protein, which results in poor mRNA–protein correlations. In addition, there are several other determinants to explain this discrepancy, e.g., post-transcriptional regulation, time lag, translational regulation, technical limitation and biological variability. All of them can more or less contribute to differences that we have observed between protein and mRNA in our experiment.

Contrary to other studies [49,51], a decline in *ICAM-1* in this set of experiments was not observed, possibly due to a different type of inflammatory challenge. While previous studies used TNF-α alone [49,52] or in combination with platelets and thrombin [51], we utilized a MIX of three different cytokines in order to better mimic the cytokine storm that endothelial cells experience during preeclampsia. However, this intense cytokine combination may have been overwhelming for the pure in vitro culture of endothelial cells, which could also potentially explain the lack of the effect of HCQ on IL-8 mRNA and secretion levels in activated CTR fpECAs observed in our study. In addition, we tested the effect of HCQ on EO-PE fpECAs. In these experiments, the focus was on IL-8 and VCAM-1 levels, because those molecules were affected in CTR fpECAs. In the case of VCAM-1, there was no difference between untreated and treated cells. However, HCQ had a deleterious effect on IL-8 secretion in EO-PE fpECAs.

Our study represents the first attempt to examine the effects of HCQ on cells derived from EO-PE patients. Notably, we observed that while a concentration of 1 µL/mL of HCQ exhibited an effect on fpECAs from healthy placentas, a ten-fold higher concentration was necessary to elicit a similar response in fpECAs isolated from EO-PE placentas. This discrepancy suggests a potentially reduced responsivity of EO-PE fpECAs that was possibly attributed to their prolonged exposure to chronic inflammation. A study by Fasano et al. demonstrated that higher HCQ levels in the whole blood of SLE patients were associated with lower levels of soluble LAM [53]. Another study in SLE patients showed that pregnant women with a low dose of HCQ in the serum had higher rates of preterm birth [54]. Further investigations are needed to better understand the underlying mechanisms and to optimize HCQ treatment strategies for EO-PE.

The strength of this study was the use of primary fpECAs isolated directly after birth from EO-PE placentas. Despite EO-PE being less common, it has been strongly associated with higher rates of neonatal mortality and a greater degree of maternal morbidity compared to late onset of PE [55,56]. The only curative option for PE is delivery of the neonate. Since EO-PE starts before 34 weeks of gestation, the approach of immediate delivery could lead to high neonatal mortality and morbidity rates and may even lead to health issues later in life [57]. Moreover, in cases of successful delivery, it would also lead to increased hospitalization time of the neonatal in intensive care unit because of prematurity. However, it is important to have a proper in vitro experimental model for screening for potential treatment options.

The present study’s limitation is a lack of a proper control for EO-PE, due to a significant difference in gestational age, neonate weight and weight percentile between CTR and EO-PE fpECAs. These differences were to be expected since early gestational age of the EO-PE group is directly related to fetal size [58]. Because international consensus defines EO-PE if occurring before 34 weeks of gestation [59], it presents a challenge to find appropriate matching healthy controls to compare to since the cause of preterm labor is inflammation [60]. Therefore, choosing CTR with preterm rupture of membranes is not appropriate either, as we studied the impact of cytokines.

Moreover, there is a need for a comparison between CTR and EO-PE fpECAs functional characteristics before and after HCQ treatment. While it is true that the behavior of CTR cells incubated with the cytokine mix could hypothetically be similar to that of EO-PE cells, we acknowledge that studying the endothelial barrier in EO-PE endothelial cells would provide additional insights. Moreover, we agree that the reaction of EO-PE cells to HCQ, even after MIX stimulation, may differ from that of CTR cells. This potential variation in response warrants further investigation and could be an important aspect to con-sider for future studies. Exploring the effects of HCQ on EO-PE endothelial cells, both in terms of molecular and functional changes, would contribute to a more comprehensive understanding of its therapeutic potential in the context of preeclampsia.

In conclusion, in this study, we show that HCQ at 1 µg/mL neither protects nor worsens endothelial functionality in CTR fpECAs, suggesting the safety of this medication. Moreover, our research has shown that in CTR fpECAs, 1 µg/mL of HCQ had a limited effect on IL-8 and VCAM-1 basal levels, while in EO-PE fpECAs, only a higher dose of HCQ was able to slightly affect the secretion of IL-8. The narrow protective effect of HCQ does not extend across the entire cytokine cascade. To the best of our knowledge, we are the first to show an impact of HCQ on fpECAs (from both CTR and EO-PE placentas) alone or in combination with cytokines. In our work, we addressed just one of many possible mechanistic cascades of PE. Therefore, to further elucidate the effect of HCQ as a possible treatment of PE, additional investigations are needed that focus on other placental cells (immune cells and trophoblasts) or different aspects of PE pathology (hypoxia, cytokine production, or relaxation of endothelium).

## 4. Materials and Methods

### 4.1. Sample Collection

The study protocol (29-319 ex 16/17) was approved by the Medical University of Graz Ethics Committee (IRB00002556), and informed consent was obtained from all patients. Pregnancies were considered CTR when there was no evidence of medical and/or obstetric complications. Preeclampsia was defined as new onset of blood pressure ≥ 140/90 mm Hg on more than two readings taken 6 h apart after 20 weeks of gestation, combined with proteinuria ≥ 300 mg/24 h. In the absence of proteinuria, preeclampsia was defined as hypertension in association with thrombocytopenia (platelet count less than 100,000/µL), impaired liver function (elevated blood levels of liver transaminases to twice the normal concentration), new development of renal insufficiency (elevated serum creatinine greater than 1.1 mg/dL or a doubling of serum creatinine in the absence of other renal disease), pulmonary edema, or new onset of cerebral or visual disturbances. In our set up, EO-PE was defined according to ACOG guidelines: when PE onset of symptoms as well as clinical manifestation occurred before 34 weeks of gestation, then it was considered EO-PE [61].

### 4.2. Preparation of Substances

HCQ sulfate in powder form was purchased from Sigma-Aldrich (H0915, Merck, Darmstadt, Germany). It was diluted with Ampuwa sterile water (Fresenius Kabi, Bad Homburg, Germany) to the concentration of 10 mg/mL. Cytokines: TNF-α (H8916, Merck, Darmstadt, Germany), IL-6 (I1395, Merck, Darmstadt, Germany) and recombinant human IL-1β (200-Q1B, PeproTech, Cranbury, NJ, USA) were diluted according to manufacturer instructions. All substances were aliquot and stored at −20 °C in dark until needed. All working dilutions were prepared with growth medium used in experiments.

### 4.3. Cell Isolation

Primary feto-placental endothelial arterial cells (fpECAs) were isolated from the chorionic plate of placentas of 14 CTR and 7 EO-PE pregnant women as previously described elsewhere [34]. They were grown on 1% porcine skin gelatin (Sigma-Aldrich, St. Louis, MO, USA) pre-coated flasks, in Endothelial Cell Growth Medium MV (PromoCell, Heidelberg, Germany) with the supplement pack provided by the supplier, 10% fetal calf serum (FCS), 0.1% gentamycin (Thermo Fisher Scientific, Waltham, MA, USA) and were used in experiments until passage 10.

### 4.4. Cell Culture

When CTR fpECAs reached 90% confluence, 10 ng/mL TNF-α (Merck, Darmstadt, Germany), 10 ng/mL IL-6 [8,62] (MIX) and 1 ng/mL IL-1β [63] in the presence or absence of 1 µg/mL HCQ (Merck, Darmstadt, Germany) were added. EO-PE fpECAs were only incubated with HCQ in two different concentrations: 1 or 10 µg/mL. It was previously shown that 1 µg/mL of HCQ represents mean concentration measured in the serum of patients who are administered the drug for systemic lupus erythematosus [64,65], while 10 µg/mL was the highest in vitro nontoxic concentration used for SARS-CoV-2 [66]. After 24 h of fpECAs incubation with HCQ, MIX or HCQ+MIX, on a 37 °C, 5% CO_2_ and 20% O_2,_ supernatant for ELISA analysis, RNA in Qiazol (Qiagen, Valencia, CA, USA) for mRNA expression and cell lysates in RIPA buffer for protein analysis were collected. Supernatants were centrifuged at 1735× *g* and 4 °C for 15 min. The collected samples were stored at −80 °C until used for experiments.

### 4.5. Wound Healing Assay

fpECAs isolated from 6 different CTR placentas were seeded in serum-deprived medium (2% FCS) in 12-well plates at a density of 10^5^ cells per well. After cells reached 100% confluence, two scratches per well were made with 100 µL pipette tips. Cells were washed with warm medium two times before fresh medium with/without HCQ and/or MIX was added. The wound-healing process was documented by Live Cell imaging with a Leica DMI 6000 B (Leica Microsystems GmbH, Wetzlar, Germany), equipped with a heating unit (37 °C) and 5% CO_2_ supply. Phase contrast microscopic photos were recorded right after scratching and at an imaging interval of 30 min for 24 h. Quantification of photos was performed with ImageJ, V 1.43l (Rasband, W.S., ImageJ, US National Institutes of Health, Bethesda, MD, USA, https://imagej.nih.gov/ij/, 1997–2018) with addition of Wound_healing_size_tool plugin [67]. The 15 h time point was taken as an end point for all fpECAs because that was the earliest time point when one CTR fpECA reaches wound closure. Scratch area was quantified in µm^2^ at 0, 5, 10 and 15 h. Further, scratch areas at each time point (5, 10 and 15 h) were normalized to 0 h scratch area and multiplied by 100 to obtain the percentage of wound area. In order to quantify healing, the percentage of wound area was subtracted from 100% (represents total closure of wound).

### 4.6. Endothelial Barrier Assay

For continuous monitoring of endothelial barrier stability, we used the xCelligence system (Roche, Basel, Switzerland). This system is based on impedance real-time sensing. Any change in cell proliferation, morphology or adherence is detected as a change in impedance by the xCelligence Real-Time Cell Analyzer, and it is reported as a dimensionless parameter called the Cell Index (CI). For endothelial barrier integrity assay, CTR fpECAs from 3 different placentas were seeded at 10^4^ cells per well in quadruplicate in 96-well E plates with golden electrodes. After 24 h, HCQ with/without MIX was added. The formation of an endothelial barrier was followed through every 15 min measurement of cell impedance. The experiment was repeated three times. Obtained data were processed in xCelligence software (v.1.2.1). Before statistical analysis, CI was normalized to the time point of substance (HCQ and MIX) addition to the cells. Comparisons between CTR and HCQ or MIX, as well as MIX vs. MIX+HCQ, were made at 5, 10 and 15 h time point.

### 4.7. Two Steps RT-qPCR

RNA extraction from all samples (CTR 9–14 and EO-PE 6–7, where exact numbers for each experiment are stated below the figures in the Results section) was performed with miRNeasy mini kit (Qiagen, Valencia, CA, USA) according to the manufacturer’s instructions. Quantification of RNA concentration and quality control were performed with QIAexpert System. Reverse transcription was performed with RT-qPCR Luna RT script (New England Biolabs, Ipswich, MA, USA) on a Thermo Cycler by manufacturer’s program recommendations. Quantification of gene expression was performed on the Real-Time PCR Detection System CFX384 Touch (Bio-rad, Hercules, CA, USA), using 5 µL of Luna RT-qPCR Universal Master Mix (New England Biolabs, Ipswich, MA, USA) together with 1 µL of QuantiTect Primer Assays (Qiagen, Valencia, CA, USA) and 4 µL of template cDNA (in concentration of (2.5 ng/µL) per well. Genes of interest whose expressions were measured were IL-6 receptor (R) 1, TNF-α R1, IL-1β R1, IL-8, ICAM-1, VCAM-1 and SELE. The run was programmed according to the manufacturer’s instructions for master mix. Samples, a no reverse transcriptase control and no template controls were amplified in triplicate. Efficiency of all used primers was also calculated, and data were corrected for it. Gene expression data were analyzed using a generalized version of the comparative threshold cycle method for relative quantification with normalization to the expression of the reference gene: Hypoxanthine phosphoribosyl transferase 1, Ribosomal Protein L30, and 18S Ribosomal RNA. All primer sequences used for qPCR analysis are listed in Appendix A.

We would like to clarify the differences in the calculation of ddCt values for our study. To perform statistical comparisons, we used dCt values. These values were obtained by subtracting the mean Ct values of three housekeeping genes from the Ct value of the gene of interest for the same sample and condition. In the first case (Figure 4), we examined the differences in basal expression between CTR and EO-PE fpECAs. The dCt of each sample was subtracted from the mean of the dCt values of all CTR fpECAs. This approach allowed us to observe potential changes in the control group as well. In other cases (Figure 5 and Figure 6), ddCt is calculated by subtracting dCt from untreated cells from each treatment within each sample. This way, ddCt of untreated cells for each sample were 0 and FC 1, so that all changes induced by different challenges (HCQ, MIX, MIX+HCQ and HCQ 1 µg/mL or 10 µg/mL) were more visible on the FC level. The differences in FC values observed between untreated CTR fpECAs, shown in Figure 4C.I,D.I and Figure 5A.I,B.I, as well as the FC values for EO-PE fpECAs, shown in Figure 4C.I,D.I and Figure 6A.1,B.I, were a result of the different experimental questions and setups used.

### 4.8. Western Blot

The total protein amount of VCAM-1 from 10 different CTR fpECAs was determined by WB assay. Dilution of primary antibody was 1:1000, while for secondary antibody, it was 1:2000. Total protein (10 µg) was separated by electrophoresis and then blotted on a nitrocellulose membrane. After 1 h blocking with 5% milk, the membrane was incubated with VCAM-1 primary antibodies (#13662, Cell Signaling Technology, Denver, MA, USA) at 4 °C overnight. Followed by 1 h washing with Triss Borat EDTE buffer with 0.1% Tween, the membrane was exposed to HRP-bonded secondary antibody for 1 h at room temperature. After another hour of washing and 5 min enhancement with SuperSignal West Pico (Bio-rad, Hercules, CA, USA), bands were visualized on Fusion FX device (Vilber, Collégien, France). As a loading control, vinculin (ab129002, Abcam, Cambridge, UK) was used. Quantification of bands was performed on Evolution Capture v17 (Fusion Software, Vilber, Collégien, France).

### 4.9. Soluble Molecules in Cell Supernatants

The changes in fpECAs’ supernatant concentration of IL-8 (N = 9) and sVCAM-1 (N = 10) after endothelial activation in the presence or absence of HCQ were measured using Human VCAM-1/CD106 Quantikine ELISA Kit and Human IL8/CXCL8 Quantikine ELISA Kit (both from R&D systems, Minneapolis, MN, USA).

### 4.10. Flow Cytometry

Flow cytometry was used for the measurement of membrane protein abundance of ICAM-1, VCAM-1 and SELE. In both assays, 2 × 10^5^ cells/well were seeded in 6 plates. Medium was exchanged for medium with/without HCQ and/or MIX the following day. After 24 h, cells were detached with TriplE (Thermo Fisher Scientific, Waltham, MA, USA). In order to block Fc-receptors and reduce non-specific binding, 100,000 cells were resuspended in 3% FCS-1xHBSS solution and incubated at room temperature for 10 min. This was followed by centrifugation at 300× *g* for 5 min. For membrane staining, cells were incubated with ICAM-1 (500 µg/mL, CD54 Pacific Blue, #322715, Biolegends, San Diego, CA, USA), VCAM-1 (100 µg/mL, CD106 APC #305809, Biolegends, San Diego, CA, USA), and SELE (100 µg/mL, CD62E PE #322605, Biolegends, San Diego, CA, USA) for 30 min at 4 °C in the dark. The next two steps were repeat washing of the cells with 2 mL of washing buffer (PBS containing 0.1% BSA and 2 mM EDTA) and 5 min centrifugation at 300× *g*. For measurements, cells were resuspended in 200 µL of washing buffer. Right before counting on a CytoFLEX flow cytometer (Beckman Coulter, Brea, CA, USA), 2 µL of 7-AAD (BD Bioscience, Franklin Lakes, NY, USA) was added to each sample for live–dead gating. Individual staining on OneComp eBeadsTM Compensation Beads (Thermo Fischer Scientific, Cat #01-1111-42) was used for compensation of surface molecules. Isotype controls corresponding to each fluorochrome in the experiment were used to detect non-specific positive signals:APC—Mouse IgG1, k isotype control, #567155 (BD Pharmingen, San Diego, CA, USA); PE—Mouse IgG2a, k isotype control, #400211 and Pacific Blue—Mouse IgG1, k, isotype control #400151 (both from BioLegends, San Diego, CA, USA).

The following gating strategy was used in order to quantify the expression of previously mentioned surface markers. Cells were separated by size using forward and size scatter (FSC-A and SSC-A). Next, doublet discrimination was performed by plotting FSC height vs. FSC area. Additional gating for 7-AAD positive and negative cells was performed, and only 10,000 7-AAD negative events were taken into account. In the last step, positive and negative surface protein cell populations were plotted in a histogram. The gating strategy for one representative sample of 9 biological repetitions is shown in Appendix A in Appendix A.

All measurements were performed on CytoFLEX flow cytometer (Beckman Coulter, Brea, CA, USA). Results were processed and quantified using FlowJoTM v.10.810 (FlowJo Single Cell Analysis Software, Ashland, OR, USA). Data are presented as median fluorescence intensity (MFI). 

### 4.11. Statistics

Statistical analyses of data were performed in GraphPad Prism v.9.5.1 (GraphPad Software, San Diego, CA, USA). All data were tested for outliers. The distribution of data (parametric vs. not parametric) was tested by the Shapiro–Wilk test (*p* ≤ 0.05, not normally distributed). When variances were not equal, Welsh correction (*t* test) or Geisser–Greenhouse (repeated measurements (RM) one-way ANOVA) correction was applied. For the comparison between two conditions, one of following tests was used: unpaired or paired two-tailed *t* test or Mann–Whitey test. Comparison between more than two groups was performed by paired RM one-way ANOVA with Sidak post hoc or Friedman test with Dunn’s post hoc.

## Figures and Tables

**Figure 1 ijms-24-10934-f001:**
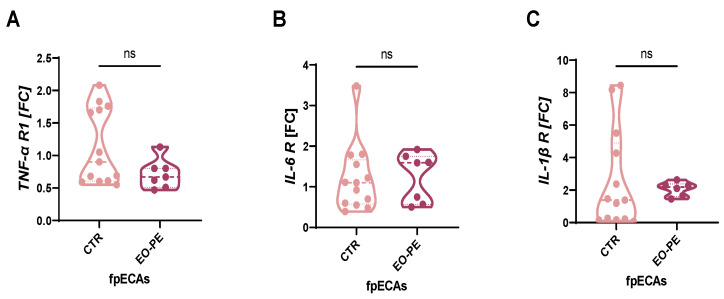
Cytokine receptors in CTR and EO-PE. Expression of TNF-α (**A**), IL-6 (**B**) and IL-1β (**C**) receptors was quantified with RT-qPCR and compared between CTR and EO-PE fpECAs. No significant difference was noted in any of them between the two sets of samples. FC of each gene was presented as a violin plot with all individual samples. Mann–Whitney test was used for comparison. Dashed lines represent median, while dotted represent quartiles. CTR N = 13, EO-PE N = 6–7. ns = not significant. RT-qPCR, quantitative reverse transcription PCR; fpECAs, feto-placental endothelial cells isolated from arteries; R, receptor; TNF-α, tumor necrosis factor α; IL, interleukin; FC, fold change.

**Figure 2 ijms-24-10934-f002:**
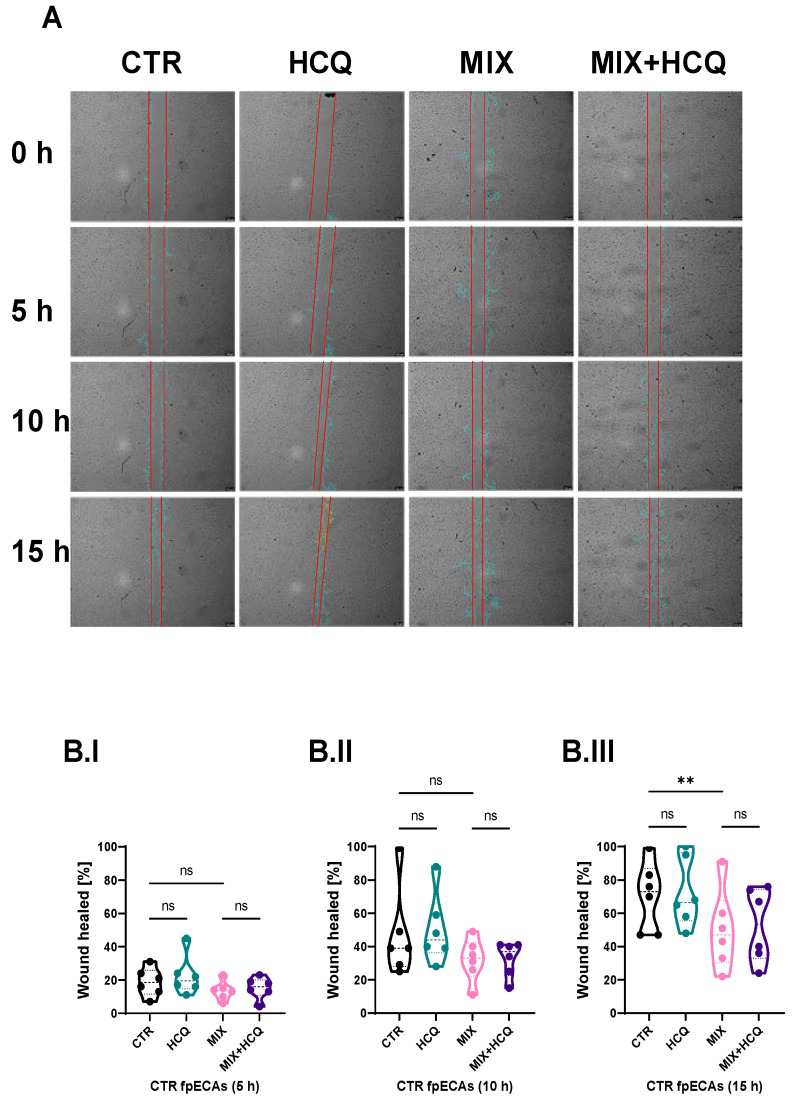
Wound healing assay. (**A**) Microscopic visualization of wound healing of one representative experiment (out of six biological repetitions) under all conditions at 0, 5, 10 and 15 h time points. Original magnification is 10×. (**B.I**–**B.III**) Statistical assessment of differences between CTR and HCQ, CTR and MIX, CTR and MIX+HCQ, and MIX and MIX+HCQ at three specific time points (5, 10 and 15 h; Quantification of wound healing was carried out using ImageJ V 1.43l software. Scratch areas of all conditions at time points 5, 10 and 15 h was normalized to respectful scratch area at time point 0 h. Repeated measurements (RM) one-way ANOVA with Sidak’s post hoc test was used to determine the significance at 5 and 15 h; Friedman test with Dunn’s post hoc was used for the 10 h time point. N = 6; ** *p* ≤ 0.01; ns, not significant. Scale bar = 200 µm; 1 µg/mL of HCQ was tested. h, hours; HCQ, hydroxychloroquine; MIX, cytokine mix composed of TNF-α, IL-6 and IL-1β.

**Figure 3 ijms-24-10934-f003:**
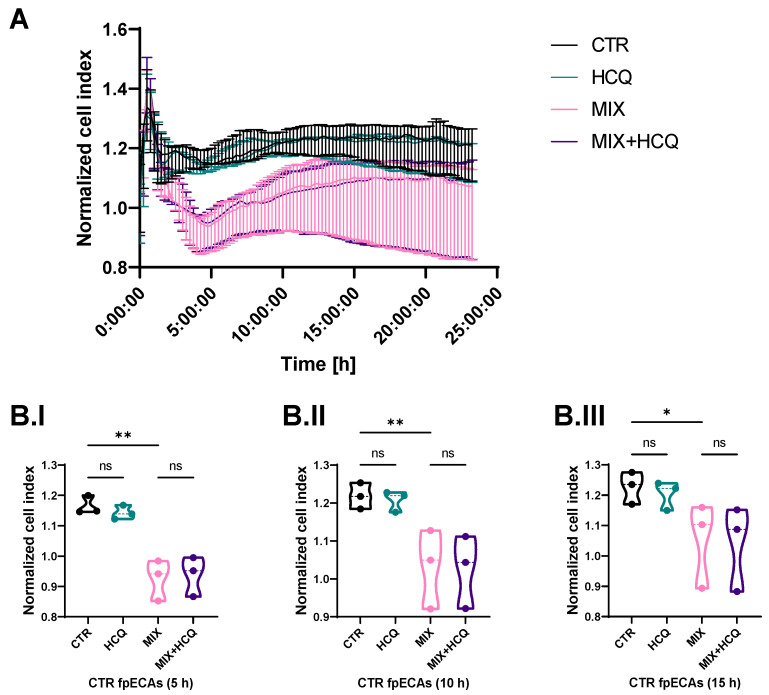
Endothelial barrier assay. (**A**) Real-time impedance measurement of endothelial barrier integrity after treatment with HCQ, MIX or combined treatment over time of CTR fpECAs from three independent experiments. Disruption of endothelial barrier was observed after challenge with MIX. (**B.I–B.III**) After quantification of cell index for specific time points, MIX treatment showed the strongest effect at 5 h (**B.I**), and a slow recovery over 10 h (**B.II**) and 15 h (**B.III**) of MIX-treated cells was detectable. Moreover, 1 µg/mL HCQ neither improved nor impaired endothelial barrier stability. Statistical analysis of differences between four conditions at specific time points (5, 10 and 15 h) was conducted with paired RM one-way ANOVA. N = 3; * *p* ≤ 0.05; ** *p* ≤ 0.01; ns, not significant.

**Figure 4 ijms-24-10934-f004:**
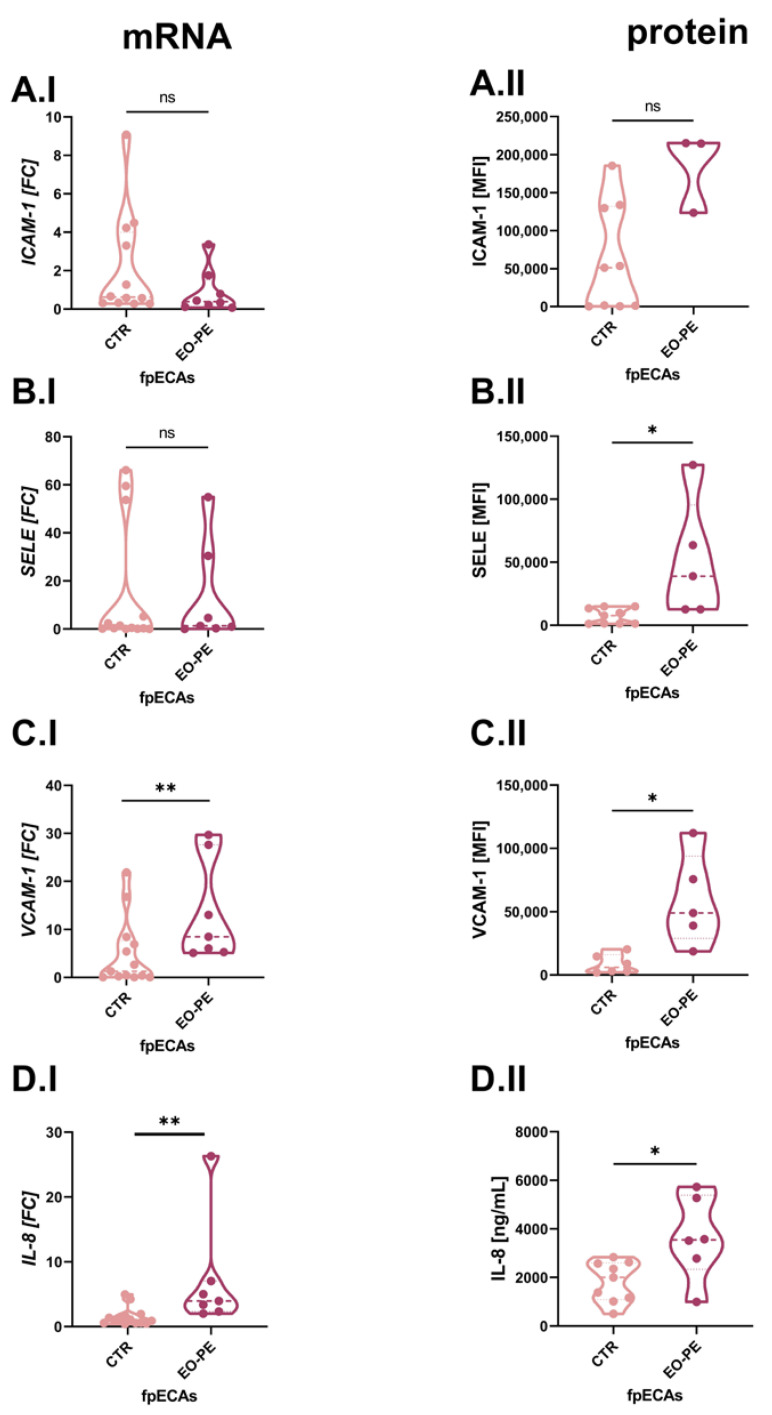
Comparison of LAM and IL-8 between CTR and EO-PE fpECAs. No significant difference in ICAM-1 was observed between CTR and EO-PE fpECAs on mRNA (**A.I**) or protein (**A.II**) levels. *SELE* (**B.I**) mRNA levels were similar between CTR and PE, but the membrane amount of SELE was higher in EO-PE (**B.II**). VCAM-1 transcription (**C.I**) was higher in EO-PE compared to CTR fpECAs. The same trend was observed for VCAM-1 membrane amount (**C.II**). EO-PE fpECAs of *IL-8* mRNA were significantly higher (**D.I**) compared to CTR fpECAs. Moreover, EO-PE fpECAs secreted (**D.II**) twice the amount of IL-8 compared to CTR fpECAs. Two-tailed unpaired *t* test with/without Welch’s correction or Mann–Whitney; CTR—N = 9–13; EO-PE—N = 6–7; * *p* ≤ 0.05; ** *p* ≤ 0.01; ns, not significant; LAM, leukocyte adhesion molecules; ICAM-1, intercellular adhesion molecule 1; SELE, selectin E; VCAM-1, vascular cell adhesion protein 1; MFI, median fluorescence intensity.

**Figure 5 ijms-24-10934-f005:**
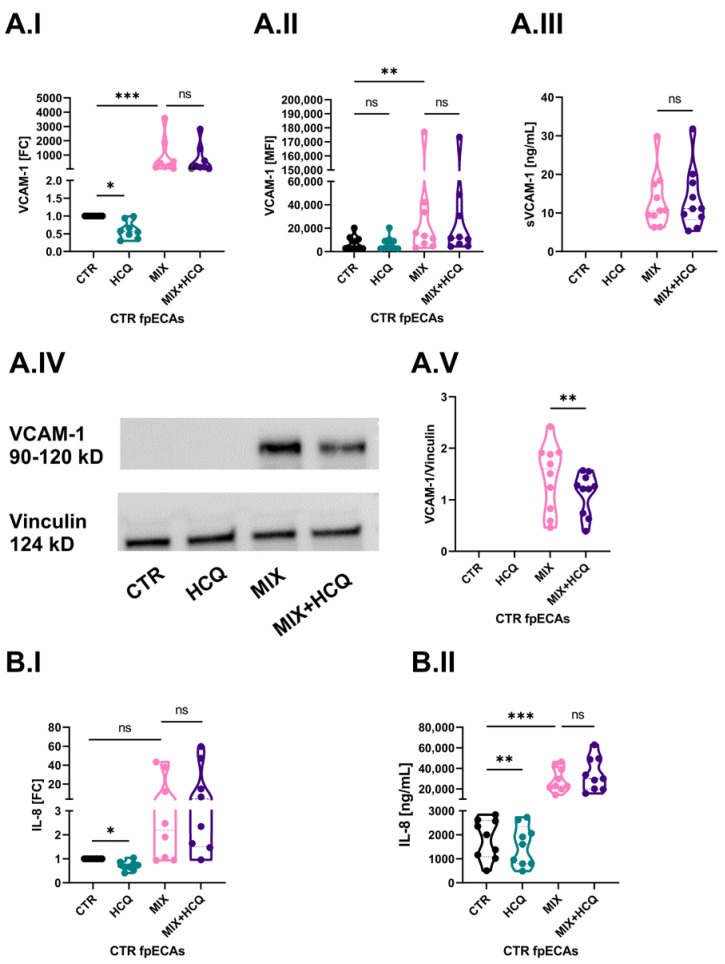
Effect of 1 µg/mL on basal and cytokine-induced levels of VCAM-1 and IL-8. (**A**) Cytokine mix activated CTR fpECAs by enlarging VCAM-1 mRNA (**A.I**) and membrane protein levels (**A.II**) compared to untreated CTR fpECAs. The secretory (**A.III**) and total protein amount (**A.IV**) of VCAM-1 measured by ELISA and WB, respectively, were detectable just after MIX challenging. HCQ alone only reduced basal mRNA amount of VCAM-1 (**A.I**), but it was unable to prevent changes led by MIX when added together. The only exception was the total protein concentration of where the addition of HCQ to MIX led to light weakening of the VCAM-1 signal (**A.V**). When this signal was quantified relative to the loading control, it was significantly lower compared to MIX challenged cells. Moreover, that decrease was not due to a change in sVCAM-1 secretion (**A.III**). (**B**) Changes in mRNA (**B.I**) and secretion (**B.II**) levels of IL-8 in CTR fpECAs were determined by RT-qPCR and ELISA. On both levels, basal levels were lower (**B.I**,**B.II**) by HCQ and were boosted by cytokine mix (**B.II**). However, HCQ was unable to prevent changes led by MIX when added together. Statistical significance for multiple comparisons was assessed using either paired RM one-way ANOVA with Sidak’s post hoc test or Friedman test with Dunn’s correction. Paired *t* test was used for the comparison of two groups. N = 8–10; * *p* ≤ 0.05; ** *p* ≤ 0.01; *** *p* ≤ 0.001; ns, not significant; WB, Western blot.

**Figure 6 ijms-24-10934-f006:**
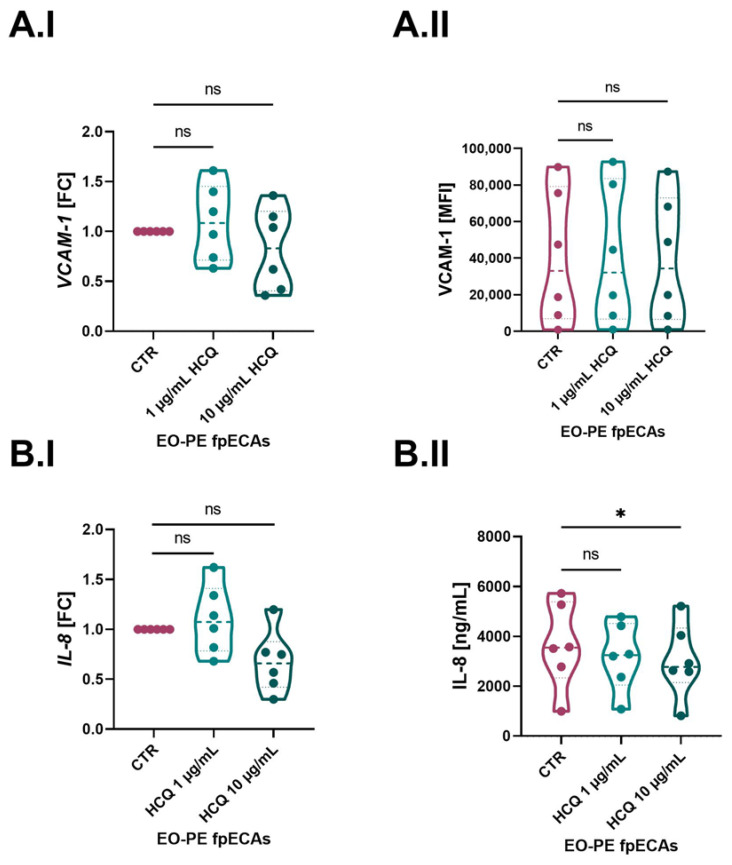
HCQ effect on IL-8 and VCAM-1 in EO-PE fpECAs. When EO-PE fpECAs were incubated with increasing dosages of HCQ, there was no difference either on mRNA grade of *VCAM-1* (**A.I**) or *IL-8* (**B.I**) as well as on the membrane presence (**A.II**) of VCAM-1 between CTR and HCQ-treated EO-PE fpECAs. In contrast, a higher dosage of HCQ (10 µg/mL) downregulated the secretion of IL-8 in EO-fpECAs (**B.II**). RM one-way ANOVA with Sidak’s post hoc was used for statistical analysis. N = 7; * *p* ≤ 0.05; ns, not significant.

**Table 1 ijms-24-10934-t001:** Characteristics of women and their offspring included in the study.

	CTR (N = 14)	EO-PE (N = 7)	*p*
**Delivery information**			
Gestational Age	38.6 ± 1.53	31.8 ± 2.33	<0.001
Mode of the delivery	SP 4/CS 10	CS 7	
**Neonatal data**			
Birth weight (g)	3490 ± 470.8	1465 ± 410	<0.001
Birth weight percentile	61.23 ± 25.93	22.83 ± 18.72	0.005
Fetal sex	7 m, 7 f	3 m, 4 f	
Umbilical cord blood Arterial, (pH)	7.26 ± 0.09	7.3 ± 0.04	0.487
Umbilical cord blood Venous, (pH)	7.35 ± 0.17	7.35 ± 0.01	0.747
**Maternal data**			
Age (years)	31.79 ± 5.74	35 ± 4.75	0.219
BMI before pregnancy (kg/m^2^)	23.81 ± 4.91	21.97 ± 5.32	0.149
Systolic blood pressure (mmHg)	125.4 ± 10.24	158 ± 9.81	<0.001
Diastolic blood pressure (mmHg)	81.54 ± 9.22	102.3 ± 8.58	<0.001
sFlt-1 (pg/mL)	/	20,768 ± 7176	
PlGF (pg/mL)	/	59.49 ± 48.35	
sFlt-1/PlGf-Ratio (pg/mL)	/	459.5 ± 241.4	

Data are represented as mean ± SD. Statistical significance was assessed by unpaired Student’s *t* test or Mann–Whitney test. CTR, control; EO-PE, early-onset preeclampsia, BMI, body mass index; SP, spontaneous delivery; CS, caesarean section; sFlt-1, soluble fms-like tyrosine kinase 1; PlGF, placental growth factor.

## Data Availability

Not applicable.

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
