# Peer review of "The Impact of Hydroxychloroquine on Primary Feto-Placental Endothelial Cells from Healthy and Early-Onset Preeclamptic Placentas"

_ijms, 2023, doi:10.3390/ijms241310934_

Round 1

Reviewer 1 Report

I find this article interesting although the title is misleading as there were minmimal effects of HCQ on the cytokine induced effects of endothelial activatation.  The article addressed for the most part the effects of HCQ on the non activated feto-placental cells (by MIX) among the control and EO-PE cells.  Although VCAM-1 was slightly reduced (non significantly) in the activated control cells by addition of HQ there appeared to be no effects on IL-8 among the control cells and no effects among the stimulated EO-PE cells with the addition of HCQ.  It is very difficult to tease out the differeces in the molecules associated with endothelial activation (IL-8 and VCAM-1) among the non-stimulated and stimulated group and also those that were incubated with and without HCQ.  These differences will need to be clarified, the title will need adjustment, and the discussion will need to emphasize the lack of major effect of HCQ with stimulation as opposed to the greater effects of HCQ on the non-stimualted cells.  I do not beleive that any portion of the study would need repeating or continuation.  If the manuscript is adjusted to reflect my recommendations then I would support publication

Acceptable

Author Response

Dear Reviewer,

Thank you for your valuable suggestions and comments on our manuscript. We appreciate your time and effort in providing constructive feedback. We have carefully considered each of your points and have made the necessary revisions to address them.

In the attached documents, you will find the answers and explanations to the questions and concerns you raised. We believe that the revisions made to the manuscript have significantly improved its clarity and addressed the raised concerns. We hope that these modifications adequately address your suggestions, and we appreciate your support for the publication of our study.

Reviewer 2 Report

This is a well-written article presenting an analysis of the potential use of HCQ in preeclampsia. They used an in vitro model to study the effects of HCQ by pursuing 3 objectives. Although no positive or negative effect of HCQ on endothelial functionality in CTR fpECAs was shown, as the authors indicate the safety of this drug. However, wouId like to ask the authors to complete the manuscript with information

- How did the authors verify that  exposing the cells to pro-inflammatory cytokines, mimic PE condition 

- discrepant results between mRNA and protein should be discussed

Author Response

(The authors gave the same response as above.)

Reviewer 3 Report

All comments are in the attached file

Author Response

(The authors gave the same response as above.)

Round 2

Reviewer 1 Report

The revisions are greatly appreciated - recommend publication